# Introducing a New Mobile Electroencephalography System and Evaluating Its Quality in Comparison to Clinical Electroencephalography

**DOI:** 10.3390/s23177440

**Published:** 2023-08-26

**Authors:** Paria Samimisabet, Laura Krieger, Themis Nethar, Gordon Pipa

**Affiliations:** Institute of Cognitive Science, Osnabrueck University, 49074 Osnabrück, Germany

**Keywords:** electroencephalogram (EEG), mobile EEG, Bluetooth, resting state, eyes open/closed

## Abstract

Electroencephalography (EEG) is a crucial tool in cognitive neuroscience, enabling the study of neurophysiological function by measuring the brain’s electrical activity. Its applications include perception, learning, memory, language, decision making and neural network mapping. Recently, interest has surged in extending EEG measurements to domestic environments. However, the high costs associated with traditional laboratory EEG systems have hindered accessibility for many individuals and researchers in education, research, and medicine. To tackle this, a mobile-EEG device named “DreamMachine” was developed. A more affordable alternative to both lab-based EEG systems and existing mobile-EEG devices. This system boasts 24 channels, 24-bit resolution, up to 6 h of battery life, portability, and a low price. Our open-source and open-hardware approach empowers cognitive neuroscience, especially in education, learning, and research, opening doors to more accessibility. This paper introduces the DreamMachine’s design and compares it with the lab-based EEG system “asalabTM” in an eyes-open and eyes-closed experiment. The Alpha band exhibited higher power in the power spectrum during eyes-closed conditions, whereas the eyes-open condition showed increased power specifically within the Delta frequency range. Our analysis confirms that the DreamMachine accurately records brain activity, meeting the necessary standards when compared to the asalabTM system.

## 1. Introduction

Electroencephalography (EEG) has emerged as a well-established and non-invasive method for assessing electrical activity within the brain [1]. Its versatile applications span across various domains, encompassing the diagnosis of brain disorders, sleep disorders, focal brain disorders, and brain mapping [2,3]. The advent of cutting-edge EEG tools has revolutionized the field by enabling real-time recording of brain activity [4]. To enhance its accessibility and overcome the constraints of cost and adaptability associated with traditional EEG systems, a new generation of portable EEG technology has been developed, known as Mobile electroencephalography (Mobile-EEG) [5]. These portable systems have paved the way for real-time analysis of brain activity, making it more convenient and efficient [6].

The introduction of a new era in EEG technology can be witnessed through the superiority of Mobile EEG systems when compared to their stationary counterparts. Not only do these systems boast a lower production cost, but they also offer enhanced portability [4,7]. Despite their potential, the current utilization of mobile solutions remains limited, predominantly confined to research settings, not reaching broader communities such as gamers, athletes, and the general population in need of medical applications or educational tools. Recognizing this challenge, an innovative concept has emerged to develop a new generation of mobile EEG systems that exhibit both acceptable quality and affordability, aiming to extend access to students, researchers, medical practitioners, and patients worldwide. With a strong emphasis on user-friendliness, this technology has been specifically designed to cater to non-professional individuals, ensuring ease of use and widespread accessibility.

The Mobile-EEG project, known as DreamMachine (Figure 1), has been designed and developed in our project [8]. This innovative device utilizes real-time electrode-transferred data, establishing seamless communication with an Android app via Bluetooth Low Energy (BLE). With a 24-bit resolution, it can accurately record and transfer brain activity data, which are then managed through the dedicated Android app. In terms of its physical appearance, DreamMachine takes the form of a rectangular device with dimensions 58.55 × 51.03 mm (L×W). Equipped with 24 channels, it includes two additional channels—one serving as a subject reference and the other as a ground. Powered by a battery, the device boasts a compact size, enabling long-term testing and transportability for tasks such as sleep and dream experiments. Notably, DreamMachine stands out for its affordability, with a manufacturing cost below 150 euros (for the time needed to design and produce the board). In the mobile EEG market, OpenBCI and Muse 2 have emerged as prominent choices, offering cost-effective solutions without compromising performance [9,10,11]. While other mobile EEG devices like MindWave Mobile2 [12], Emotiv EPOC+ [13], and the unicorn hybrid black [14] are recognized for their advanced features and higher price points, Muse 2 and OpenBCI provide more accessible alternatives [15,16]. However, it is important to acknowledge the limitations of these devices, including a limited number of channels—16 channels for the Cyton+Daisy board from OpenBCI [17] and 4 channels for the Muse 2 [11]. A higher cost of $1999.00 USD for the OpenBCI system and 249 USD for the Muse 2, along with potential challenges in terms of accessibility, which may pose constraints for certain student and research budgets, are noted.

In response to the aforementioned challenges, the DreamMachine was developed as a solution to enhance the accessibility and affordability of mobile EEG technology. One of the other goals of this device is to provide a greater number of channels in comparison to other low-cost devices which are cited in the literature review of the latest studies on wireless EEG systems [18], while simultaneously ensuring cost-effectiveness. A notable obstacle associated with OpenBCI systems is the need for technical expertise in their setup and operation, which poses difficulties for users lacking familiarity with hardware and software integration. In contrast, the DreamMachine was designed with a user-friendly approach, featuring an intuitive design and interface to simplify the overall user experience. Availability is another challenge encountered with OpenBCI, Muse 2, and other commercial devices, which can hinder their widespread adoption and impede research endeavors. To surmount this constraint, the DreamMachine places considerable focus on enhancing accessibility and availability by implementing open-source and open-hardware initiatives. Additionally, OpenBCI devices tend to have larger form factors and limited portability, restricting their suitability for certain research applications. In contrast, the DreamMachine prioritizes compactness and portability, enabling EEG experiments to be conducted in diverse environments beyond the confines of traditional laboratory settings. One notable advantage of the DreamMachine is its comprehensive 24-channel configuration, enabling users to capture more detailed and precise brainwave activity compared to the 16 channels offered by OpenBCI and only 4 channels by Muse 2. Notably, the DreamMachine’s resolution and battery life rival those of systems like Muse2. The DreamMachine maintains 24-bit resolution and up to 6 h of battery life, aligning with or exceeding the specifications of other systems, specifically, in comparison with the Muse2 system, which provides only 12-bit resolution and 5 h of battery life [11]. By leveraging the advantages of OpenBCI and Muse 2, the DreamMachine endeavors to further reduce costs, offer more channels, higher resolution, longer battery life, and customizable hardware to suit EEG-specific design requirements, while also incorporating widely used Bluetooth connectivity. This integration enables users to record EEG data via Android systems even in non-laboratory environments.

The initial part of this paper focuses on presenting the DreamMachine system, specifically examining its hardware board design and firmware details. In the Method section, the system architecture of DreamMachine will be dissected, with its components, circuit design, and interplay between digital and analog elements being meticulously broken down. This examination will provide a comprehensive understanding of the employed components and how the seamless communication between digital and analog aspects contributes to the system’s effectiveness. Within the Data Acquisition part, a careful comparison between the DreamMachine and the ANT system will be conducted to comprehend their performance in different scenarios. The investigation’s focal point encompasses an eyes-open and eyes-closed experiment, functioning as a rigorous test to compare the performance of our novel DreamMachine against the ANT system. The selected methodology for conducting these experiments and the subsequent signal-processing steps will be thoroughly explained, offering insights into our approach. Moving to the Results section, the outcomes of these experiments will be presented, displaying the processed signals from both the DreamMachine and the ANT system. Subsequently, in the Discussion section, the results will be carefully examined to comprehend their implications. This analysis aids in understanding how well our innovative mobile EEG system performs compared to standard EEG systems. These efforts contribute to the broader discourse on the advancement of EEG technology.

## 2. Methods

Prior to delving into the specifics of hardware circuit design and communication aspects, the establishment of a comprehensive understanding of the overall architecture of the DreamMachine system becomes imperative. Within this section, the examination of the system’s architecture is initiated, with the intricate interplay among its diverse components being detailed. Furthermore, the meticulous exploration of the hardware board circuit design takes place, shedding light on the identification of components and their arrangement. Additionally, the investigation of the communication process encompasses the transition from analog to digital conversion, succeeded by the method implemented for the seamless transfer of data to the Bluetooth chip within the DreamMachine.

### 2.1. System Architecture

Figure 2 illustrates the block diagram depicting the system architecture. The system comprises three analog-to-digital converters (ADC), a Bluetooth low-energy (BLE) transmitter device and a novel Android application named EEGDroid. The primary purpose of EEGDroid is to record and store EEG signals within its interface. Additionally, EEGDroid offers various functionalities for interacting with the DreamMachine system configuration. Subsequent sections will delve into the intricate specifics of the hardware board design and firmware implementation for this innovative device.

### 2.2. Board Design

The board schematic of the DreamMachine device is shown in Figure 3. The design is divided into three sections: Analog, Digital, and Power supply components.

In the core of the Analog area of Figure 3, three AD7779ACPZ ADCs components are demonstrated. These components are specifically chosen for their ability to capture high-resolution EEG and ECG signals [19]. Each ADC provides eight channels with a 24-bit simultaneous sampling ADC. By combining three ADCs, each with eight channels, the DreamMachine achieves a total of 24 recording channels. The Analog Inputs (AIN+0 to AIN+7) pins of the three ADCs were connected to 24 channels. Channels 1–8 were connected to the pins of ADC1, channels 9–16 to ADC2, and channels 17–24 to ADC3. All AIN-0 to AIN-7 pins of the ADCs were connected to the active reference electrode, making the DreamMachine a common reference EEG system. In addition, an extra regulator was utilized for the ground. Because the AD7779ACPZ contains a 2.5 V reference, the subject ground was connected to half of the reference ground, which is 1.25 V. Therefore, this extra channel acts as a ground to prevent power line noise from interfering with the small biopotential signals.

The Digital section entailed a BC832 chip and Serial Peripheral Interface (SPI) instructions to enable serial communication with all three ADCs. The BC832 chip handles the pre-processing of the sampled EEG signal and facilitates its transmission via BLE. The Firmware section provides more details about Bluetooth performance and how it transfers data.

Two independent regulators for the digital and analog components are utilized in the Power Supply sector. Voltage Common Collector (VCC) Analog and Digital serve as the power suppliers for all circuits and components within the respective digital and analog segments. All circuitry is battery powered, using the high capacity of 3.7 V (650 mAh) lithium-polymer battery. Furthermore, the prototype DreamMachine consumes a maximum of 110 mA with all channels engaged, resulting in a total recording time of up to approximately 6 h when running on batteries. This can be enhanced using an external power bank connected via a micro-USB.

### 2.3. Firmware

The BLE chip was connected to the three ADCs using the Serial Port Interface (SPI) and an additional connection known as the Data Output Ready (DRDY) pin, which indicates the completion of signal conversion. The SPI interface was used to configure the ADCs and read the conversion results, while the BLE chip works as the SPI interface master device.

The ADCs were set up to generate samples at a rate of 500 Hz. In Figure 4, the process of the internal data pipeline is described. The ADCs were configured to generate samples at 500 Hz. After receiving a signal from the DRDY pin, the conversion result is actively pulled by the BLE chip from each ADCs. The obtained samples undergo a series of processing steps. Initially, the samples were filtered using a low-pass filter and a notch Infinite Impulse Response (IIR) filter. Subsequently, three samples from 3 ADCs are internally averaged, resulting in the downsampling of the ADC signal from 500 Hz to either 250 Hz or 167 Hz. Finally, an IIR or first-order high-pass filter is applied to further refine the signal. After downsampling and processing the measurements for each channel, they are bundled together into a BLE packet representing a single data point. Each BLE packet consists of a 1-byte header followed by the channel values. The header contains a 4-bit packet ID, and an additional four bits encode the number of internally dropped packets since the last transmission (e.g., due to full BLE buffers). The channel values are not encoded in their entirety (24-bit) to save bandwidth, but rather as the difference between the current and previous measurements which were encoded as a channel value. The channel values are signed integers with lengths of 10, 14, or 16 bits. They are aligned within the packets without any spacing, allowing values to span byte boundaries. The DreamMachine device advertises a service with UUID 0x0EEG, which encompasses five characteristics as outlined in Table 1.

In standby mode, when no notifications are enabled, the DreamMachine device remains idle and disregards the DRDY signal. During this state, configuration adjustments can be made through the 0xECC0-characteristic, which is further described in the subsequent section. Once notifications are enabled for the 0xC0DE and one of the 0xEEGx-characteristics, the DreamMachine device initiates the measurement process and transmits channel values via the corresponding 0xEEGx-characteristic, adhering to the configured bit length. Since only 10–16 bits are transferred at this level, the difference values are appropriately clipped to fit within the specified limits.

To capture the relevant information accurately, the DreamMachine device employs a floating average mechanism to track significant changes. This allows for adaptability in different amplitude phases, where it is crucial to monitor substantial fluctuations during high-amplitude periods and capture intricate details during low-amplitude phases such as EEG waves. For this reason, the floating average was tracked to determine the most relevant measurements. Each second, this average was used to determine which part of the 24-bit difference values was the most pertinent. The bit shift undertaken to extract the relevant section is communicated via the 0xC0DE characteristic as 4-bit values for each channel. Ultimately, the output data containing the relevant measurements are transmitted via Bluetooth to the connected Android application.

The DreamMachine system can be seamlessly controlled and configured through the dedicated Android App. This application offers a range of robust features. Noteworthy functionalities include the ability to zoom in or out on signals, turning on or off specific channels display, selecting from four distinct gain settings, adjusting the sampling frequency rate with options of 250 Hz or 167 Hz, and choosing from various filtering options. Moreover, the Android App provides users with real-time signal-monitoring capabilities, allowing them to inspect the signals and assess their quality visually. This feature enables users to make informed decisions regarding the configuration settings that best suit their specific tasks or requirements. Additionally, it is important to note that the recorded signal format when using EEGDroid is in the .csv file format.

Based on the findings of the conducted study, the default filtering configuration in the novel EEGDroid application has been carefully set to encompass the best-performing specifications [8]. This ensures optimal signal processing and enhances the overall functionality and usability of the DreamMachine system. Figure 5 is being incorporated to depict the signal quality obtained from the DreamMachine before the initiation of experimental procedures. This figure is intended to showcase a 3–4 s duration of the acquired signal, thereby offering insight into the baseline quality of the collected data.

## 3. Data Acquisition

A comparative analysis was conducted between the DreamMachine device and the clinical EEG system known as asalab™ (ANT Neuro) for recording EEG signals. Both recordings utilized the waveguard™ original ANT Neuro EEG cap to ensure consistency. The DreamMachine device was configured with default settings, including a notch filter ranging from 46 Hz to 54 Hz (order 4), a gain of 8, and a low-pass filter set at 60 Hz (order 6). The EEG signals were sampled using a 16-bit A/D converter at a frequency of 250 Hz. For the clinical EEG device, a sampling frequency of 250 Hz was selected. Data acquisition was performed using 24 electrodes placed on the scalp of a healthy 23-year-old male subject. The electrode placement followed the 10–20 standard system, with the electrodes positioned at FP1, FPZ, FP2, F7, F3, FZ, F4, F8, M1, T7, C3, CZ, C4, T8, M2, P7, P3, PZ, P4, P8, POZ, O1, OZ, and O2 [20].

During the experiment, the subject maintained a distance of 90 cm from the screen, and the research protocol received approval from the Ethics Committee for the Protection of Human Subjects. All electrodes were referenced to the left Mastoid, while the right Mastoid served as the ground reference. To ensure optimal signal quality, the impedance of all electrodes was kept below 5 kΩ.

### 3.1. Methodology

Data were collected from a single participant using two distinct EEG devices, namely the DreamMachine and a standard clinical EEG system. The primary objective was to ensure comparability between the two devices, thereby maintaining consistency in the experimental conditions. The duration of the testing experience was 20 min, during which the participant underwent alternating periods of eyes open and eyes closed conditions. Each condition lasted for 2 min, with the initial segment starting with eyes closed. The participant received specific training to focus their gaze on a fixation cross displayed on a monitor throughout the task, ensuring minimal movement.

To facilitate the concurrent recording of data of the DreamMachine and triggers, a connection was established between the Android device and the laptop. This connection involved utilizing a cable to connect the two devices while ensuring that the Wi-Fi functionality was turned off on both devices. Additionally, the “Tethering” feature on the Android device was enabled. This configuration allowed for seamless and synchronized recording of data from the DreamMachine and the triggers using the LabRecorder App [21]. The participant was instructed to use the arrow keys for the eyes-open (top arrow) condition and the eyes-closed condition (down arrow). After repeating the experiment for both EEG systems, subsequent processing and analysis of the recorded data were performed using the same procedures. In this project, the Fieldtrip toolbox was utilized for data analysis [22].

### 3.2. Data Extraction

In the preprocessing phase, the loaded data underwent several steps, including applying a bandpass filter ranging from 1 to 30 Hz, performing baseline correction, and removing bad channels. The data were then divided into two groups: eyes open and eyes closed. Each group was further segmented into 15 s intervals. Subsequently, bad epochs were identified and removed from the cleaned eyes-open and eyes-closed data sets. The data sets were analyzed using the multi-taper method with Fast Fourier Transform (FFT) to compute power spectra. The data were examined in various frequency bands to gain insights into the behavior of brain activity under different conditions. It this project, the frequency bands are selected as Delta (1–4 Hz), Theta (4–8 Hz), Alpha (8–12 Hz), and Beta (12–30 Hz). Finally, a cluster-based permutation test was employed to compare the power spectral density between the ‘open’ and ‘closed’ conditions using EEG data. This approach facilitated a more comprehensive analysis of the data. The “Results” section provides a detailed explanation of the findings derived from the data analysis process.

## 4. Results

This section delves into a comprehensive analysis of 24-channel EEG data recorded using two distinct devices: the DreamMachine and the standard EEG system, asalab™ (ANT Neuro), examining both the time and frequency domains. The visual representation of the data is depicted through four figures, with Figure 6A illustrating the eyes-open condition for both EEG systems, while Figure 6B portrays the eyes-closed condition for the same devices.

In Figure 6A, the brain activity captured by the DreamMachine is displayed on the right side, while the raw data from the ANT system (standard EEG system) are showcased on the left side. It is evident that both devices exhibit a low-amplitude signal in the eyes-open condition, along with noticeable eye blinking artifacts (shown by a red dashed rectangle). This similarity in results between the two devices is promising, indicating the effectiveness of the DreamMachine in capturing relevant brain activity during this state. 

Moving on to Figure 6B, the eyes-closed condition is presented, and once again, the DreamMachine’s brain activity data are juxtaposed with the raw data from the ANT system. In this case, both devices display the presence of a distinguishable alpha wave which is highlighted by a red dashed rectangle. This concurrence in results reinforces the reliability of the DreamMachine in detecting and recording alpha activity during the eyes-closed state.

Importantly, it should be noted that the raw data presented in both figures are prior to any artifact removal processes. This approach aims to highlight the genuine brain activity captured by the DreamMachine system.

### Data Performance Analysis of DreamMachine

DreamMachine performance: The performance of the DreamMachine was evaluated through Figure 7, which depicts the raw EEG time series captured during two distinct conditions: eyes closed (A) and eyes open after triggering the subject (B and C). In all 24 channels, a low-amplitude signal in the eyes open condition with eye movement (B) and eye blinking (C) artifacts is recognizable. Just after opening the eyes (B), an eye movement oscillation peaking with high amplitude and frequency is displayed and known as Saccades, compared to the signal behavior before the trigger. In the eyes-closed state (A), the Alpha rhythm (8–12 Hz) exhibited heightened activity in the occipital region’s channels (O2, OZ, O1, POZ), which are responsible for processing visual information. Alpha oscillations are believed to be linked to the inhibition of visual processing during rest periods [23]. Conversely, in the eyes-open condition, the Delta (1.5–3.5 Hz) rhythm was primarily triggered in the prefrontal cortex (FP1, PF2, FPZ) as well as the frontal region (F7, F3, F8, F4, and Fz). This finding indicates the engagement of specific brain regions associated with attention and cognitive processes when the eyes are open [24].

Up until now, the raw data from both devices in the time domain have been depicted without applying filtering, the same as what users could observe with real-time data through the EEGDroid application. Subsequently, the clean data are analyzed using the Morlet multitaper method with Fast Fourier Transform (FFT), and the results of this analysis, along with the statistical findings, are presented. The power spectral density and spectrogram analyses were performed in the frequency domain throughout the 24 channels under both conditions [25]. In this analysis, seven channels, namely FP1, FP2, FPz, POz, O1, O2, and OZ as the most effective channels were selected. The signals of DreamMachine and the standard EEG system regarding the power spectral density using a Hanning window are analyzed. The result of the cluster-based permutation test of all selected channels is shown in Figure 8A using the Standard EEG device. The topographical plots (‘eyes closed minus eyes open’ and ‘eyes open minus eyes closed’) visually display the statistically significant differences in power spectral density. Clusters of electrodes with significant differences are indicated by color-coded regions, where lower *p*-values (*p* < 0.01) correspond to more intense colors. These results showed that the higher power spectral density is activated, particularly in the occipital region, whenever the subject has closed their eyes.

In contrast, the frontal region of the brain is more activated when the eyelids are open. Additionally, increased Delta activity was anticipated regarding the expected behavior in the eyes-open condition, reflecting heightened frontal activation and cognitive processing. In contrast, during eyes-closed conditions, an increase in alpha activity was expected, signifying a state of relaxation or drowsiness. Figure 8B demonstrated the same analysis from the DreamMachine device. The results indicated that both devices exhibited similar patterns in terms of power spectral density during eyes-open and eyes-closed conditions. This similarity in spectral characteristics suggests that both devices captured comparable neural activity patterns for their respective conditions. The topographical analysis of the power spectral density confirmed consistent patterns across the devices, reinforcing their reliability and validity for recording EEG data in eyes-open and eyes-closed conditions.

Figure 9A,B illustrate the power spectrum density of the standard EEG and the DreamMachine signals, respectively. The spectrogram of channel Oz is presented on the top side, showcasing data recorded by both devices. During eyes closed conditions, a higher power density was observed at approximately 10 Hz, indicative of increased Alpha frequency activity. Conversely, when the subject’s eyes were open, a higher power density at approximately 1.5–3.5 Hz, associated with the Delta rhythm, was detected. This underscores the occurrence of distinct brain activity patterns depending on the eyes’ status (open or closed). In the lower part of the Figures, the power spectrum density of the average of seven channels is displayed for both conditions. The difference in mean and standard deviation demonstrates a noticeable increase in power density at 10 Hz for both devices. However, eyes-open conditions exhibited a significantly suppressed response compared to eyes-closed conditions. To mitigate the interference caused by power line frequency noise (AC), a notch filter was employed, resulting in a dramatic reduction in power at the 50 Hz frequency range. This noise, if present in EEG recordings, can impede the detection and analysis of neural activity of interest [26]. These findings highlight the distinctive neural activity patterns in the eyes-closed and eyes-open conditions, emphasizing greater activation in the occipital region during eyes-closed states, and enhanced frontal region activity during eyes-open conditions was performed for both systems to assess the statistical significance of the differences in power spectral density between the ‘open’ and ‘closed’ conditions at specific electrode locations, including FP1, FP2, Pz, POz, O2, O1, and Oz. The resulting topographical plots display the clusters of statistically significant differences, with the magnitude and direction of these effects represented by a color map. Regions with intense colors on the topographical plots indicate the presence of significant differences, while uniform color patterns suggest no significant effects at those electrodes. This analysis enables the identification of the precise locations where the ‘open’ and ‘closed’ conditions exhibit significant discrepancies in power spectral density.

## 5. Discussion

This paper introduces an innovative mobile EEG device, the DreamMachine, which boasts user-friendliness, enhanced accessibility for EEG recording, and data collection. Its cost-effectiveness, Bluetooth and battery-driven features, as well as its compact size, unlock new possibilities, including the potential to collect data from up to 1000 subjects in home environments. Moreover, the device enables remote monitoring and testing for patients with conditions like epilepsy, who require assessments from afar. Furthermore, we conduct a comparative analysis of the device’s signal quality with a standard EEG system through eyes-closed and eyes-open experiments. The results of these experiments are detailed in the following section, shedding light on the performance of both systems. 

Alpha wave: In the time domain, a notable increase in amplitude was observed following the eyes-closed trigger, indicating a higher frequency occurrence after eyelid closure. In the frequency domain, the difference between the two conditions was more pronounced, as illustrated in Figure 9A. The spectrum analysis of the standard EEG system revealed a significant increase in power around 10 Hz during eyes-closed compared to eyes-open conditions. Similarly, the DreamMachine system demonstrated a similar behavior, as depicted in Figure 9B.

Delta wave: The Delta band (1.5–3.5 Hz) is typically associated with deep sleep and unconsciousness [27]. However, recent research indicates that this band can also manifest during wakefulness, particularly when the eyes are open, [28]. This phenomenon is known as ”wakeful Delta activity” and has been observed across different age groups, including children, adults, and older adults [29]. Analysis of frequency domain figures demonstrates that the Delta power band, ranging from 1 to 3.5 Hz, exhibits greater power during eyes-open conditions in both systems. This indicates the presence of heightened Delta activity during eyes-open conditions compared to eyes-closed conditions. This pattern is also apparent in the time domain, consistent with previous studies [30].

Within the landscape of wireless EEG systems, a significant role is played by the DreamMachine, characterized by a harmonious fusion of features. Emerging as a robust choice for intricate research and applications within brain-computer interfaces, comprehensive coverage is facilitated through its 24 EEG channels. The system’s precision in capturing intricate brain dynamics is emphasized through its 250 Hz sampling rate and 24-bit resolution. DreamMachine strikes a balance between battery life and affordability, ensuring 6 h of sustained operation without compromising usability. Remarkably, these features are offered at an approximate cost of $163.17 USD (less than 150 €), rendering it an economically feasible solution for both research professionals and personal brain activity enthusiasts. As shown in Table 2, within the context of a comparative analysis derived from the recent literature review [18], DreamMachine distinctly emerges with a competitive edge:

EPOCX (2020) is presented with 14 channels and a high 2048 Hz sampling rate; however, its system fee of $849 USD is placed within a higher cost bracket. In contrast, DreamMachine has achieved affordability while maintaining a comparable channel count and sampling rate. Unicorn Hybrid Black (2019) is surpassed by DreamMachine in both channel count and battery life. With 24 channels and 6 h of operational endurance, DreamMachine is standing out, catering to exhaustive data collection and prolonged usage scenarios. Muse2 (2018) is exhibiting evident advantages in DreamMachine’s higher channel count, longer battery life, and cost-effectiveness. With 24 channels compared to Muse2’s 4 channels, DreamMachine is achieving a wider coverage. Additionally, DreamMachine’s 6 h battery life is surpassing Muse2’s 5 h, while remaining marginally higher in cost. MindWave Mobile 2 (2018) is showcasing DreamMachine’s superiority through its comprehensive specifications. With 1 channel and 8 h of battery life, MindWave Mobile 2 is overshadowed by DreamMachine’s 24 channels and 6 h of battery life. Moreover, DreamMachine is presenting better value owing to its competitive cost. OpenBCI Cyton + Daisy EEG System (2015) is outperformed by DreamMachine in channel count, sampling rate, and cost-efficiency. The former’s 16 channels and 125 Hz sampling rate are surpassed by DreamMachine’s 24 channels and 250 Hz sampling rate, ensuring heightened accuracy in signal acquisition.

In summary, DreamMachine’s highly advantageous profile is highlighted. Its comprehensive attributes, coupled with affordability and technological integration, emphasize its appeal when contrasted with other systems. The equilibrium between channel count, sampling rate, battery life, open access, and cost efficiency underscores DreamMachine’s potency across a wide spectrum of applications, spanning advanced research to personal brain activity monitoring.

## 6. Conclusions

This study introduced DreamMachine, a Mobile EEG system, and conducted a comparative evaluation by comparing it with a standard clinical EEG system using the same subject in an experimental setting. The experiment involved recording the eyes open/closed resting state using both devices. The analysis revealed significant differences in the power spectrum of the Alpha band between the eyes-closed and eyes-open conditions for both devices. Specifically, the power spectrum was notably higher in the eyes-closed condition. Similarly, in the case of the Delta frequency, the power spectrum was observed to increase during eyes-open conditions for both devices.

While these initial findings shed light on the performance of DreamMachine in comparison to a standard clinical EEG system, further research in this domain is warranted. Particularly, more extensive data collection is essential to gain a comprehensive understanding of the interaction between mobile and cloud-based devices and services. Such additional data will facilitate the optimization and harmonization of the infrastructure supporting mobile and cloud devices and services.

## Figures and Tables

**Figure 1 sensors-23-07440-f001:**
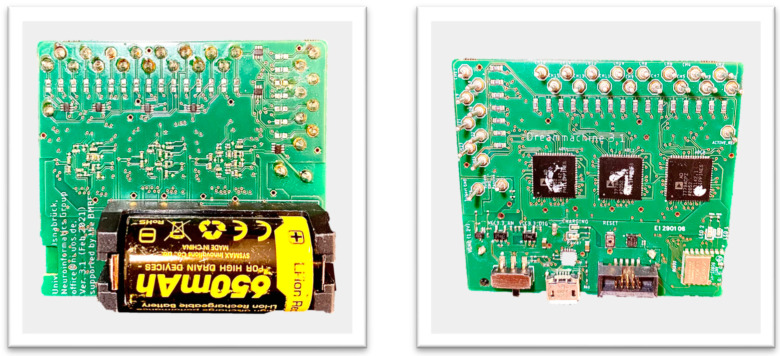
Illustration of the DreamMachine Hardware System. The left side displays the top view of the device, while the right side showcases the bottom view of the system.

**Figure 2 sensors-23-07440-f002:**
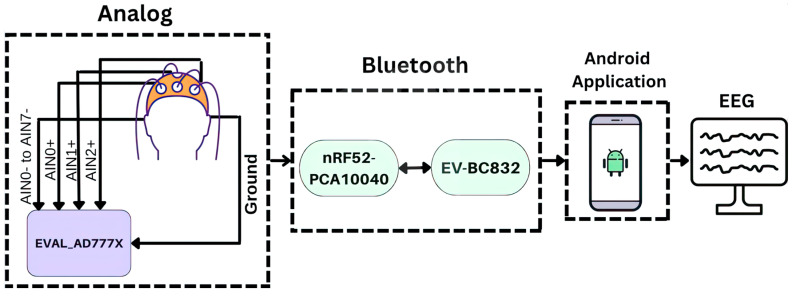
The Mobile-EEG system layout.

**Figure 3 sensors-23-07440-f003:**
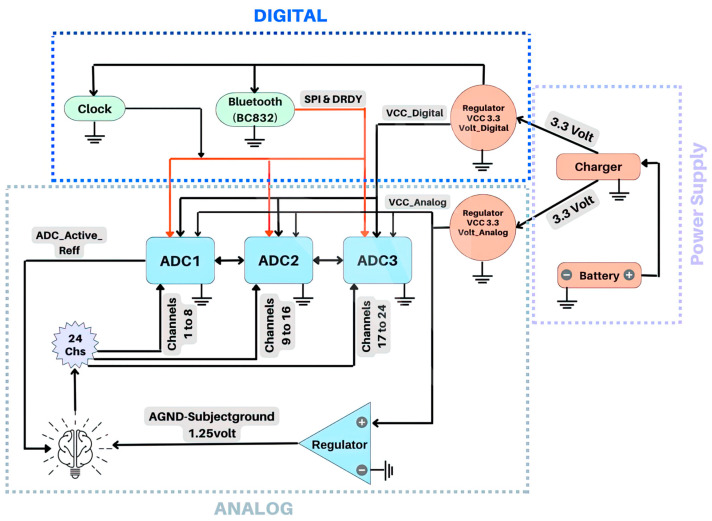
Board schematic design of the DreamMachine device.

**Figure 4 sensors-23-07440-f004:**
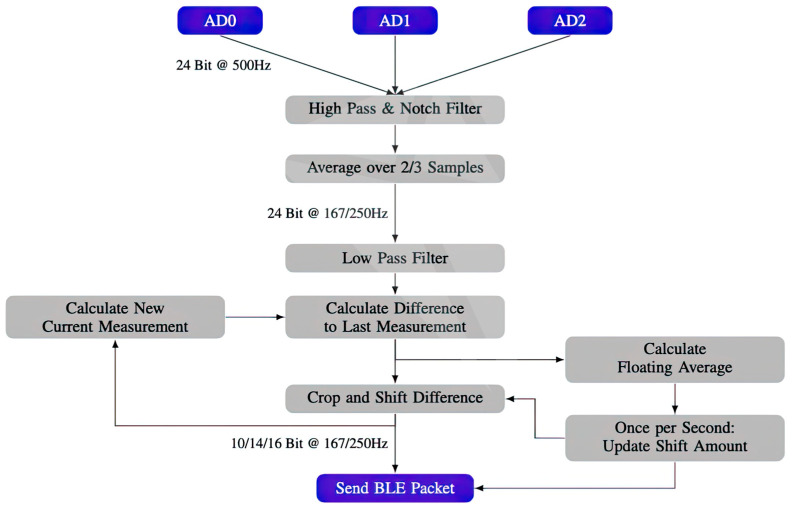
Visualization of the data pipeline, including bit widths and sampling frequencies.

**Figure 5 sensors-23-07440-f005:**
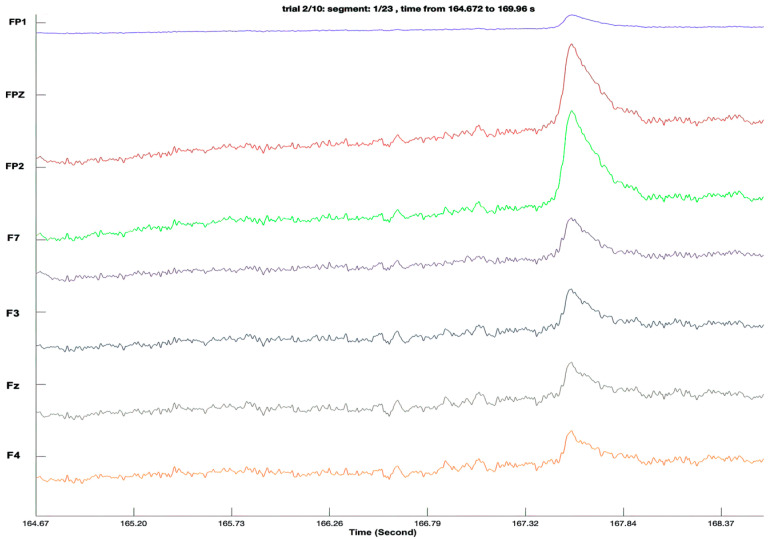
Signal Quality Assessment: The figure displays a representative snippet of the signal acquired from the DreamMachine. The showcased 3–4 s interval serves as a preliminary demonstration of the data quality, offering a glimpse into the fidelity and clarity of the recorded EEG signal.

**Figure 6 sensors-23-07440-f006:**
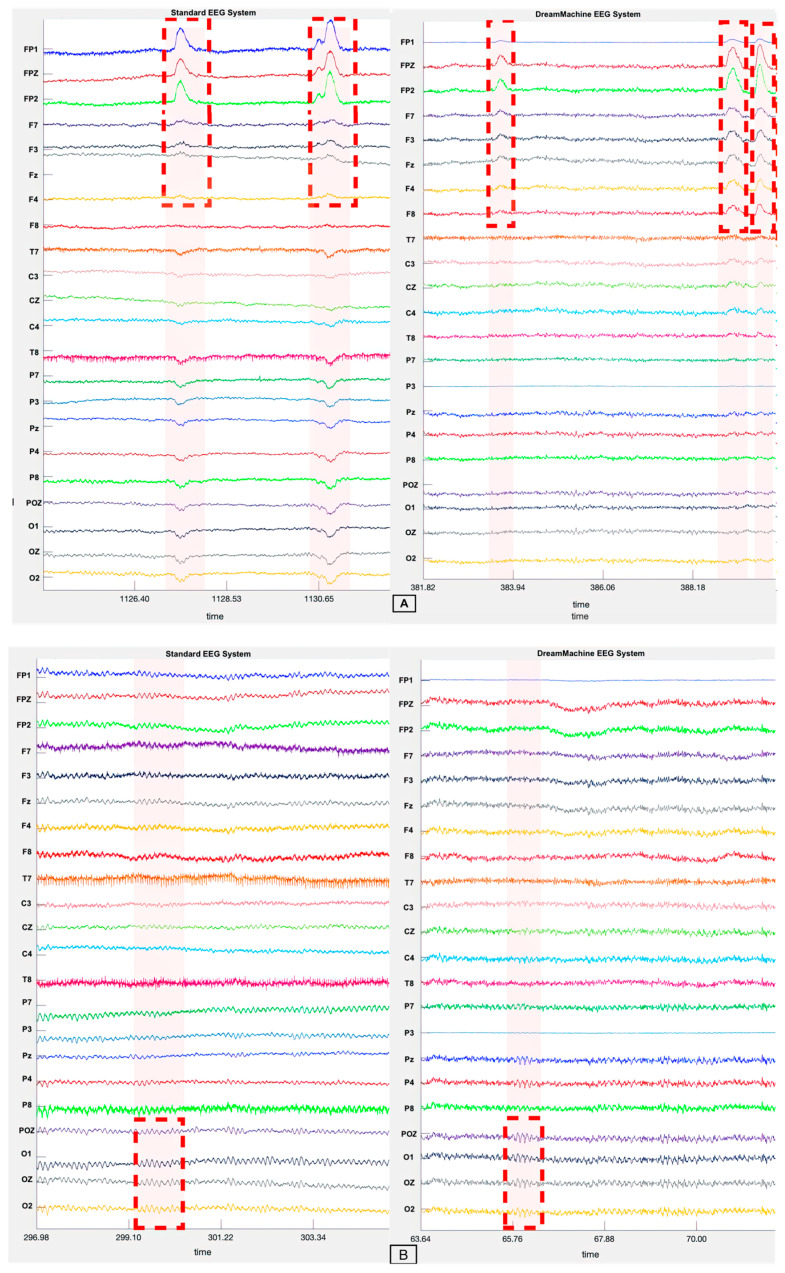
(**A**) The EEG traces from both EEG systems during the eyes-open condition reveal specific artifacts related to eye blinking. These artifacts are highlighted by red dashed rectangles and occur when the subject blinks while their eyes are open during the resting state. Notably, the artifacts are more pronounced in the FP1, FP2, and Fpz channels compared to the rest of the channels. (**B**) EEG traces of both EEG systems in eyes-closed condition. In this figure presentation, the presence of an alpha wave pattern is revealed in four channels situated within the occipital lobe. The alpha wave is visually emphasized using red dashed rectangles, and the characteristic waveform is distinctly demonstrated by the four channels, namely POz, O1, Oz, and O2.

**Figure 7 sensors-23-07440-f007:**
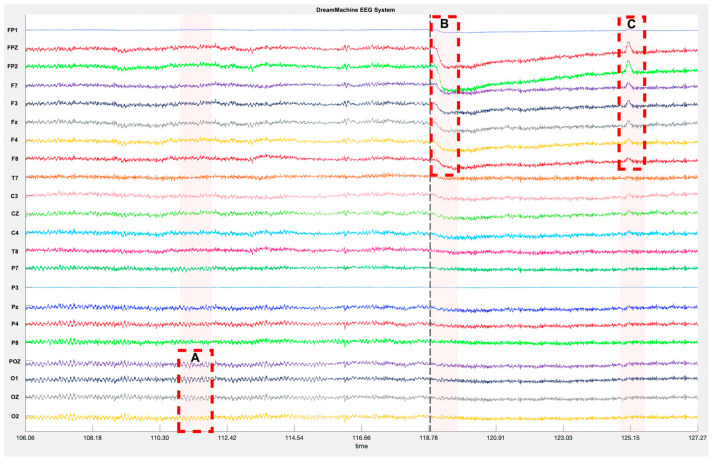
Showcases raw EEG time series recordings in two distinct conditions with DreamMachine system: eyes closed (A) and eyes open after triggering the subject (B and C).

**Figure 8 sensors-23-07440-f008:**
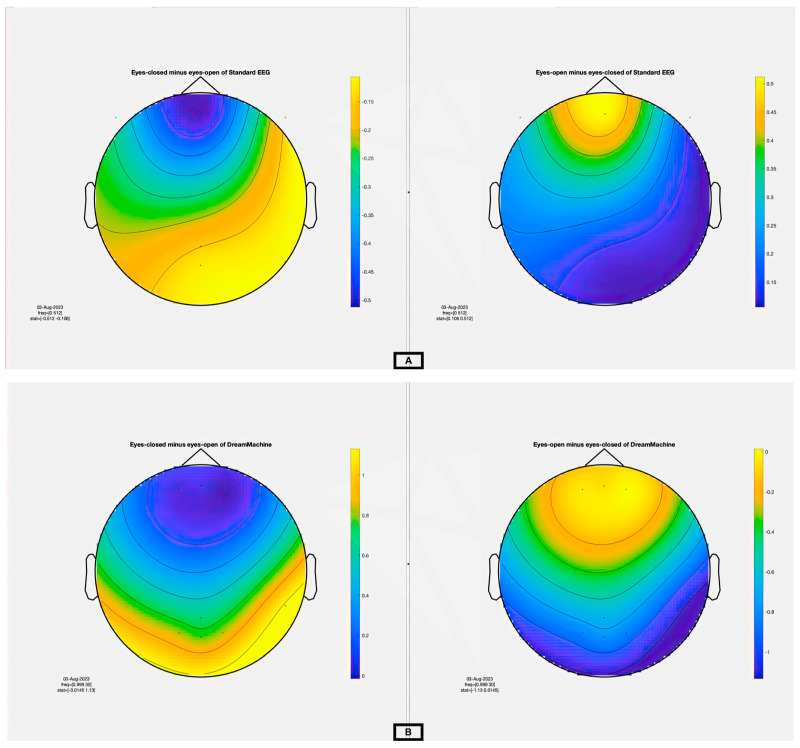
A comprehensive statistical analysis using the cluster-based permutation test. (**A**) The right side presents the result of eyes open minus eyes closed, and the left side shows the difference between eyes closed and eyes open for the Standard EEG system. (**B**) The right side presents the result of eyes open minus eyes closed, and the left side shows the difference between eyes closed and eyes open for the DreamMachine system.

**Figure 9 sensors-23-07440-f009:**
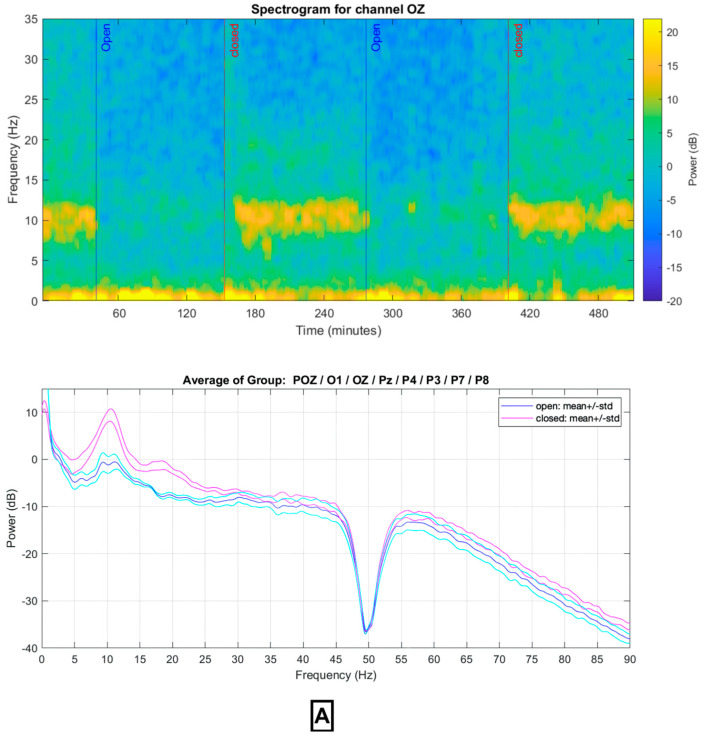
Spectrogram and Power Spectrum of (**A**) Standard EEG System and (**B**) DreamMachine System. At the top of the figure, we present the spectrogram of channel Oz for both the DreamMachine and Standard EEG systems. As evident, both devices exhibit heightened power between 8 and 12 Hz during eyes−closed conditions. Moving to the lower section of the figure, we depict the contrast and aggregate of mean and standard deviation values between eyes−open and eyes−closed states. Notably, a distinct peak emerges at 10 Hz for both devices during eyes−closed conditions, indicating enhanced Alpha activity. The prominent dip at 50 Hz signifies the presence of a notch filter effectively attenuating the 50 Hz household alternating current (AC) interference.

**Table 1 sensors-23-07440-t001:** BLE characteristics offered by the DreamMachine.

Purpose	Size [Byte]	UUID
channel values 10-bit length	31	0xEE60
channel values 14-bit length	43	0xEE61
channel values 16-bit length	49	0xEE62
configuration and battery readback	8	0xECC0
encoding updates	14	0xC0DE

**Table 2 sensors-23-07440-t002:** Specifications and Information of Wireless EEG Systems.

Devices	Channels	Sampling Rate (Hz)	Resolution (Bits)	Battery Life (Hours)	System Fee (USD)	Announcement
DreamMachine	24	250	24	6	~163.17	2023
EPOC^X^	14	2048	14/16	9	849	2020
Unicorn hybrid black	8	250	24	2	~1076.92	2019
Muse2	4	256	12	5	249.99	2018
MindWav Mobile 2	1	512	12	8	~217.45	2018
OpenBCI Cyton + Daisy EEG System	16	125	24	Depends to the Configuration	1999.00	2015

## Data Availability

Not applicable.

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
