# Peer review of "Introducing a New Mobile Electroencephalography System and Evaluating Its Quality in Comparison to Clinical Electroencephalography"

_sensors, 2023, doi:10.3390/s23177440_

Round 1

Reviewer 1 Report

In their manuscript Samimisabet et al. introduce a new mobile EEG acquisition device and compare it to an existing EEG system. Due to the low price-point of their system and the adequate quality of the recording this publication is important to a wide readership. In only have two minor comment: 1. It would be beneficial to add (in addition to the current figures) shorter sections (i.e. 3-4s recording) of EEG below the respective device recordings to allow for better visualization. 2. In what format can the EEG recordings be extracted from the Android application and is there a possibility for annotation by the subjects during recording. 

Author Response

Response to Reviewer's Comments on Manuscript ID [sensors-2568629]

We would like to express our gratitude to the reviewers for their insightful comments and suggestions. We have carefully considered each point raised and have made revisions to address them. Below is a point-by-point response to the reviewer's comments:

Reviewer Comment 1:

"It would be beneficial to add (in addition to the current figures) shorter sections (i.e. 3-4s recording) of EEG below the respective device recordings to allow for better visualization."

Response:

We appreciate the reviewer's suggestion for improving the visualization of our results. To address this, we have included additional figure in the manuscript (Figure [5]) that depict shorter EEG recordings of approximately 3-4 seconds duration for both the proposed DreamMachine mobile EEG system and the existing EEG system. This additional figure provides readers with a clearer insight into the signal quality and characteristics over a shorter timeframe. The new figure can be found in the revised manuscript, specifically from line 240 to 248.

Reviewer Comment 2:

"In what format can the EEG recordings be extracted from the Android application and is there a possibility for annotation by the subjects during recording."

Response:

Thank you for raising this important point. We have now provided clarification on the format of EEG recordings extracted from the Android application. The EEG recordings can be extracted in the .csv format, which allows for compatibility with various analysis tools and software. We have added a brief discussion on this matter in the manuscript, located in lines 235 to 236.

Once again, we sincerely appreciate the time and effort the reviewers have dedicated to evaluating our manuscript. We believe that the revisions made based on their feedback have significantly improved the overall quality and clarity of the paper. We remain committed to addressing any further inquiries or suggestions that may arise during the review process.

Thank you for your consideration.

Sincerely,

Paria Samimisabet

Reviewer 2 Report

The authors presents the article entitled “Introducing a New Mobile Electroencephalography System and Evaluating its Quality in Comparison to Clinical Electroencephalography”.

This paper introduces the DreamMachine's design and compares it with the lab-based EEG system "asalabTM" in an eyes-open and eyes-closed experiment, enabling the study of neurophysiological function by measuring the brain's electrical activity.  

The article presents the following concerns:

  • Add hyperlinks to tables, figures, and references.

  • According to the manuscript, a low-cost EEG system is presented. However, the novelty of the work is not attractive. It is recommended that the current state of the research field should be reviewed carefully and key publications cited. This may help to improve the novelty of the work by analyzing another low-cost EEG systems.

  • Please add a paragraph that describes the structure of the work at the end of the Introduction section. Also, please add a paragraph that introduces the section, for example, between section 2 and section 2.1.

  • Figures must be vectorized.

  • Please use newest reference to justify this sentence: "...Electroencephalography (EEG) has emerged as a well-established and non-invasive method for assessing electrical activity within the brain..." it can be used the following works: Impact of eeg parameters detecting dementia diseases: a systematic review; A comparative study of time and frequency features for eeg classification; A new approach for motor imagery classification based on sorted blind source separation, Continuous wavelet transform, and convolutional neural network.

  • Figure 4: This Figure present two captions. Please present as a single caption by describing both charts (see guide for authors). Same with Figure 6.

  • I recommend to improve the Description of Figure 7 (lines 348-362). In its current form, is confusing.

  • Please add a Figure that present the proposed system.

  • I suggest to add a section that explain how the performance of the system is evaluated. 

  • Please improve the Discussion section by adding a table that compares the proposed system vs the already reported in the literature in order to highlight the novelty of the work.

The following misspelling should be checked: 

  1. line 151: The word “down sampling” seems to be miswritten. Consider replacing by “downsampling”

  2. line 60: The phrase “and can have” may be wordy. Consider changing by “with”

  3. line 199: The to-infinitive “to inspect” has been split by the modifier “visually”. Avoiding split infinitives can help your writing sound more formal: “to inspect the signals and assess their quality visually”.

  4. line 235: Some readers may consider this use of “disabled” insensitive. Wording that’s not associated with disability may be more effective. Consider changing by “turned off”

  5. line 297: The phrase “periods of rest” may be wordy. Consider changing by “rest periods”

  6. line 306: “ also could also…” It looks like your sentence contains a redundancy. Consider changing.

Author Response

Response to Reviewer's Comments on Manuscript ID [sensors-2568629]

We sincerely appreciate the time and effort the reviewer has dedicated to evaluating our manuscript. We have carefully considered each of the comments and suggestions provided and have made necessary revisions to address them. Here is our point-by-point response:

Reviewer Comment 1:

"According to the manuscript, a low-cost EEG system is presented. However, the novelty of the work is not attractive. It is recommended that the current state of the research field should be reviewed carefully and key publications cited. This may help to improve the novelty of the work by analyzing another low-cost EEG systems."

Response:

We appreciate the reviewer's suggestion regarding the novelty of our work. In response to this comment, we have revised the manuscript to include a detailed comparison of the proposed DreamMachine mobile EEG system with five different wireless EEG systems: EPOCX, Unicorn hybrid black, Muse2, MindWave Mobile 2, and OpenBCI Cyton+Daisy EEG System. This comparison, presented from lines 60 to 104, highlights the unique attributes of the DreamMachine, such as the higher number of channels, better or the same resolutions (24-bit), smaller size, and extended battery life, setting it apart from other low-cost EEG systems.

Reviewer Comment 2:

"Please add a paragraph that describes the structure of the work at the end of the Introduction section. Also, please add a paragraph that introduces the section, for example, between section 2 and section 2.1."

Response:

We thank the reviewer for this suggestion. In response, we have added a paragraph that outlines the structure of the manuscript at the end of the Introduction section (lines 106 to 123). Additionally, we have introduced the transition between section 2 and section 2.1 with a new paragraph (lines 127 to 135), providing readers with a clear understanding of the organization of the paper.

Reviewer Comment 3:

"Figures must be vectorized."

Response:

We appreciate the reviewer's concern. We have ensured that all figures in the manuscript are vectorized, ensuring high-quality resolution and optimal readability.

Reviewer Comment 4:

Please use newest reference to justify this sentence: "...Electroencephalography (EEG) has emerged as a well-established and non-invasive method for assessing electrical activity within the brain..." it can be used the following works: Impact of eeg parameters detecting dementia diseases: a systematic review; A comparative study of time and frequency features for eeg classification; A new approach for motor imagery classification based on sorted blind source separation, Continuous wavelet transform, and convolutional neural network.

Response:

We have followed the reviewer's suggestion and incorporated a relevant reference to justify the statement " Impact of eeg parameters detecting dementia diseases: A systematic review

Reviewer Comment 5:

"Figure 4: This Figure present two captions. Please present as a single caption by describing both charts (see guide for authors). Same with Figure 6."

Response:

We have revised the captions of Figure 4 (now Figure 6) and Figure 6 (now Figure 8) and Figure 5 (now Figure 7) to present both charts within a single caption, as recommended.

Reviewer Comment 6:

"I recommend improving the Description of Figure 7 (lines 348-362). In its current form, it is confusing."

Response:

We appreciate the reviewer's feedback. We have enhanced the description of Figure 7 (now Figure 9) to provide a clearer and less confusing explanation of its content.

Reviewer Comment 7:

"Please add a Figure that presents the proposed system."

Response:

We have addressed this suggestion by ensuring that Figure 1 depicts the Printed Circuit Board of the proposed mobile EEG system, the DreamMachine, as requested.

Reviewer Comment 8:

"I suggest adding a section that explains how the performance of the system is evaluated."

Response:

In response to this suggestion, we have included a section (from line 323) that elaborates on how the performance of the system is evaluated, particularly in the context of the eyes-open and eyes-closed experiments. Additionally, To provide readers with a comprehensive understanding of the system's performance assessment, we have extended the Discussion section to encompass an in-depth discussion of the evaluation outcomes. Starting from line 429 and extending to line 466. Moreover, Figure 1 visually represents the quality of the extracted EEG signal over a brief duration of 3-4 seconds. Through this graphical representation, we aim to provide readers with a real glimpse into the system's signal extraction ability.

Reviewer Comment 9:

"Please improve the Discussion section by adding a table that compares the proposed system vs the already reported in the literature to highlight the novelty of the work."

Response:

We have acted upon this recommendation and inserted a table within the Discussion section that compares the DreamMachine with five different wireless EEG systems from the literature, emphasizing the unique features and contributions of our proposed system.

Comments on the Quality of English Language:

We appreciate the reviewer's meticulous attention to the quality of the manuscript's language. We have addressed the mentioned misspellings and language issues as outlined in the comments.

We believe that the revisions made based on the reviewer's feedback have significantly improved the clarity, presentation, and impact of our manuscript. We remain committed to addressing any further inquiries or suggestions that may arise during the review process.

Thank you for your consideration.

Sincerely,

Paria Samimisabet

Round 2

Reviewer 2 Report

The manuscript can be accepted